# Marine Gel Interactions with Hydrophilic and Hydrophobic Pollutants

**DOI:** 10.3390/gels7030083

**Published:** 2021-07-06

**Authors:** Peter H. Santschi, Wei-Chun Chin, Antonietta Quigg, Chen Xu, Manoj Kamalanathan, Peng Lin, Ruei-Feng Shiu

**Affiliations:** 1Department of Marine and Coastal Environmental Science, Texas A&M University at Galveston, Galveston, TX 77554, USA; xuc@tamug.edu (C.X.); pengl1104@tamug.edu (P.L.); 2Department of Bioengineering, University of California, Merced, CA 95343, USA; wchin2@ucmerced.edu; 3Department of Marine Biology, Texas A&M University at Galveston, Galveston, TX 77554, USA; quigga@tamug.edu (A.Q.); manojka@tamug.edu (M.K.); 4Institute of Marine Environment and Ecology, National Taiwan Ocean University, Keelung 20224, Taiwan; rfshiu@mail.ntou.edu.tw; 5Center of Excellence for the Oceans, National Taiwan Ocean University, Keelung 20224, Taiwan

**Keywords:** marine gels, aggregates, marine snow, hydrophobic and hydrophilic interactions

## Abstract

Microgels play critical roles in a variety of processes in the ocean, including element cycling, particle interactions, microbial ecology, food web dynamics, air–sea exchange, and pollutant distribution and transport. Exopolymeric substances (EPS) from various marine microbes are one of the major sources for marine microgels. Due to their amphiphilic nature, many types of pollutants, especially hydrophobic ones, have been found to preferentially associate with marine microgels. The interactions between pollutants and microgels can significantly impact the transport, sedimentation, distribution, and the ultimate fate of these pollutants in the ocean. This review on marine gels focuses on the discussion of the interactions between gel-forming EPS and pollutants, such as oil and other hydrophobic pollutants, nanoparticles, and metal ions.

## 1. Introduction

Pollutants in the environment encompass many extraneous substances that, when interacting with natural organic matter (NOM), change their properties as they then become parts of a new, macromolecular, complex. Pollutants are mostly human-made and include hydrophilic metal ions, hydrophobic or amphiphilic low-molecular-weight organic molecules, and nanoparticles, including micro- and nano-plastics. Very often then, these pollutants are ‘hitching’ a ride with the natural organic molecules, which are composed of terrestrially derived humic and fulvic substances, and microbially secreted EPS. While most of the literature on interactions between metal ions and NOM is devoted to understanding the binding strength, and the extent and kinetics of binding, there is much less known on the nonspecific interactions of metal ions with gel-forming EPS that can modify its gel properties. In this paper, we focus on reviewing the recent literature on interactions between gel-forming EPS and pollutants such as oil and other hydrophobic pollutants, nanoparticles, and metal ions.

EPS are mainly composed of proteins and polysaccharides, as well as smaller amounts of nucleic acids, lipids, and humic substances. EPS make up an important part of NOM in the ocean, in its particulate, colloidal, and macromolecular forms [1]. The plankton–EPS system is a dynamic system, whereby phytoplankton and bacteria form a synergistic relationship in the phycosphere. Phytoplankton secrete photosynthesized carbohydrates and polysaccharides, and associated bacteria degrade some of this material and make available other compounds such as vitamin B12 [2] and hydroxamate siderophores [3,4] to phytoplankton.

The microbial community can regulate the physico-chemical properties of the released EPS in response to changing conditions by secreting [5] polysaccharide-rich EPS (mostly phytoplankton) and protein-rich EPS (mostly bacteria [6]). These biopolymers can interact and bond with each other via ionic forces, van der Waals forces, electrostatic forces, hydrophobic interactions, hydrogen linkages, and crosslinking through chemical bonds. In EPS gels, all these forces can be active, depending on the chemical composition, e.g., proteins vs. polysaccharides. In Table 1, the terminology used in this paper is summarized.

Transparent exopolymeric particles, TEP, are commonly considered precursors of EPS [7,8,9]. They are ubiquitously present in marine and fresh water systems yet ‘nonvisible’ under the microscope unless they are stained (e.g., Alcian blue; [10]). TEP are primarily assessed as acidic polysaccharides [7]. EPS and TEP do not refer to exactly the same materials: TEP are exopolymers, but not all exopolymeric substances occur as TEP or can form TEP ([7]). EPS forming microbial biofilms have shown to be gels [11,12,13,14,15,16,17,18,19,20,21]. However, TEP are not strictly considered to be gels, as their formation relies on coagulation theory, not on intermolecular energies and assembly processes as for gel formation. Nevertheless, these terms (EPS, TEP, and gels) are often used interchangeably, as in the case of biofilm formation and biofouling [12].

Gels are conceptually considered a type of soft matter [13] and are well-defined [14]. However, microgel concentrations are operationally determined using flow cytometry, after staining with chlortetracycline, expressed as total organic carbon concentration [15], with the kinetics of gel formation determined using dynamic light scattering over hours to days [14]. Gels can also be visualized using environmental electron microscopy [14,16] and/or confocal laser scanning microscopy (e.g., [16]). Coomassie stainable particles (CSP), which are protein-containing particles and can be stained with Coomassie Brilliant Blue, are another type of gel-like particles, proposed by Long and Azam [17], that have been identified in seawater, freshwater, and phytoplankton cultures. TEP and CSP could be discrete particles, or subunits of the same particles [18].

As stated above, EPS are not a defined chemical compound, and their size is in the nano- to micro-size range. TEP are commonly assessed operationally by assaying using the Alcian blue staining of particles collected on a 0.7 µm filter [13,14,15,20,21]. As has been demonstrated by Hung et al. [22], this method can be biased, but it is nonetheless widely used. EPS are commonly assessed by the sum of the major components, proteins, and polysaccharides of a colloidal or particulate sample [23,24,25]. Gels, on the other hand, are assessed by flow cytometry, electron microscopy, or dynamic light scattering (DLS) in the filter-passing fraction [14]. Xu et al. [26] were the first to inter-calibrate the three methods, and they found reasonable agreement between them. Before proteins and carbohydrates can be assessed in the filter-passing fraction, EPS have to be pre-concentrated using ultrafiltration, dialysis, or similar techniques. Analytical methods that have been used to determine the major components of EPS include spectrophotometric assays, FTIR, Raman, GC-MS, HPLC, electron microscopy, and NMR. Although proteins and polysaccharides are determined separately, they mostly co-exist in the same macromolecules such as proteoglycans or glycoproteins [9]. Carbohydrates and proteins are determined spectrophotochemically as monomers produced in a sample after a hydrolysis step, and are calibrated against standards, while individual monosaccharides or amino acids can also be determined by HPLC [27]. On the other hand, both polysaccharides and proteins can be more quantitatively determined by NMR and FTIR, as no digestion step is needed [26].

The physico-chemical behavior of EPS (e.g., attachment and aggregation) is mostly determined by the relative hydrophobicity of EPS. Proteins, because of their amphiphilic nature, are considered to contribute most to the relative hydrophobicity of EPS. Their net charge, and thus, their relative hydrophilicity, is dependent on the ambient pH. Amino acids that have hydrophobic side chains are glycine (Gly), alanine (Ala), valine (Val), leucine (Leu), isoleucine (Ile), proline (Pro), phenylalanine (Phe), methionine (Met), and tryptophan (Trp). Individual sugars have different relative hydrophilicities, e.g., pentoses are usually less hydrophilic than hexoses, which is related to the CH-surface area of sugar molecules accessible to water molecules [28].

Proteins are important for the initial attachment process to surfaces [29]. Proteinaceous components of the biofilm matrix include secreted extracellular proteins, cell surface adhesins, and protein subunits of cell appendages such as flagella and pili [30]. Proteins also stabilize the biofilm matrix and three-dimensional biofilm architecture, while proteinaceous enzymes are involved in the degradation of the biofilm components. 

The ratio of proteins to carbohydrates of EPS (P/C) has been found to be closely related to the ‘stickiness’ of EPS and their relative hydrophobicity. For example, the P/C ratio is related to aggregation propensity, e.g., [20], surface tension [21], presence of nano-plastics or oil in microbial cultures [19,31], light-induced chemical crosslinking [23], and, when mineral matter is present, the sedimentation efficiency of marine snow [24]. Figure 1 shows some examples of how these properties can be related to the P/C ratio. Furthermore, the hydraulic residence time or sedimentation efficiency in wastewater treatment systems is also related to the P/C ratio [25]. Protein/carbohydrate ratios of EPS aggregates are thus an indicator of attachment propensity, i.e., its ‘stickiness’ [32], which can also be directly assessed by magnetic tweezers [33]. Compared with the laborious chemical techniques needed to directly measure protein and carbohydrate content, the P/C ratio can also be expediently obtained with simple fluorescence measurements [31]. The P/C ratio can be a more convenient and informative parameter for the assessment of EPS aggregation behaviors.

All these physical properties depend on various physical and chemical factors, such as cross-linker density, cross-liner types, polymer length, pH, types of polymers, temperature, degree of swelling, or temperature. Unfortunately, to the best of our knowledge, there is no available direct measurement for these properties for natural EPS gels in the literature. However, several studies on alginate (or other purified EPS) are available in the literature that might provide some rough assessments. Mechanical and rheological (viscoelastic) properties of alginate gels were shown to be dependent on the cross-linker type, density, ionic conditions, gelling temperatures, or EPS concentrations [36,37,38,39]. The specific gravity of a typical synthetic hydrogel (PVA (polyvinyl alcohol) gels) is around 1.05 [40]. For EPS sludges, the density has been reported to be around 1.004–1.048 (g/mL) [41]. For alginate gels depending on the gelling conditions, the value can vary from 1.03 to 1.12 (g/mL) [36]. However, these specific gravity measurements were conducted in non-seawater conditions (in lower salt conditions). Please note that the specific gravity of seawater is around 1.025. The specific gravity of EPS gels in seawater might shift from the measurements in non-seawater environments. The appearance and sizes of marine EPS gels are highly heterogeneous. No typical or characteristic morphology or shape has been found or concluded. The size of EPS gels in seawater can range from sub-micrometers to millimeters, even several centimeters.

EPS are highly heterogeneous mixtures of biopolymers from various microbes in seawater and are generally associated with different types of particles (anthropogenic, minerals, or biological debris), and these physical properties (rheology, morphology, or specific gravity) of natural EPS gels are usually complex and highly variable to determine or generalize.

## 2. Relative Hydrophobicity of EPS

Exudates from different aquatic organisms can have hydrophilic and hydrophobic moieties. Mostly hydrophilic exudates include the so-called hydrocolloids that are secreted by macro-algae, e.g., seaweeds, and harvested for their distinct chemical properties valued in the food industry as thickening and gelling agents (e.g., [42]). They include acid polysaccharides such as alginates, carrageenans, pectins, gums, and more neutral polysaccharides such as agar and similar substances, extracted from seaweeds, bacteria, and other organisms [42]. Most of these, but not all of them, form gels in the presence of metal ions such as Ca^2+^. The kind and location of acid functional group determines their food or physiological properties, e.g., alginates are blood coagulants, while carrageenans are anti-coagulants [43].

Due to the solubility limitation of water, hydrophobic moieties of EPS are normally not exposed to the water but, rather, are found in the interior of the structure or proteins or humic substances. As a consequence, EPS and humic substances become amphiphilic. The relative hydrophobicity of these biomolecules, represented by the hydrophobic contact area (HCA), is an important parameter that regulates the kinetics and extent of particle aggregation and dis-aggregation reactions in the water column, and thus influences the removal of associated radionuclides (e.g., Thorium-234) and organic pollutants (e.g., petroleum hydrocarbons). Xu et al. [20] found that the P/C ratio of EPS, determined by FTIR, is linearly related to the HCA, determined by HPLC. This implies that the P/C ratio can be used as an indicator for the relative hydrophobicity of macromolecules. This then also implies that the relative hydrophobicity of the carrier biopolymers of pollutants is mainly controlled by their relative protein contents [34]. High protein content in EPS has also been found to greatly accelerate the formation of marine gels that are not subject to disaggregation after EDTA addition that complexes the Ca^2+^ that are bridging hydrophilic components of EPS, thus rendering gel formation through irreversible hydrophobic interactions [28,44,45].

## 3. Stability of Microgels upon the Addition of Amphiphiles, e.g., Dispersants

Contrary to the results of [46], which showed the instability of marine gels when irradiated by UV, there is now ample evidence that sunlight irradiation causes reactive oxygen species (ROS)-mediated chemical crosslinking reactions, leading to the photoflocculation of NOM [47]. This was shown through increases in the concentrations of molecular weight, particle size, and mass [23]. On the other hand, global change-induced increases in temperature and hydrogen ion concentrations will have the tendency to decrease the stability of gels [48].

In addition, the aggregation and dispersion of marine gels can be affected by heterogeneous particles and agents in surrounding environments. For example, nano-carbonaceous particles were shown to reduce marine gel formation, due to their negative surface charges interfering with Ca^2+^ bridge cross-linking [49]. This observation is consistent with Zhang et al. [50], which suggests that quantum dots with negative charges have a stronger capability to stabilize EPS gel than positively charged ones. In addition, the microgel size significantly decreased when in the presence of surfactants, especially in the anionic type. Furthermore, negatively charged surfactants such as sodium dodecyl sulfate (SDS) can disrupt existing native microgels, converting larger aggregates into smaller particles. Notably, in addition to human-made pollutants, the input of natural substances can also cause changes in the dynamics of microgels. Shiu et al. [51] indicated that the self-assembly of marine gels would be decreased in the presence of NOM such as Suwannee River humic acid, fulvic acid, and natural organic matter at low concentrations (0.1–10 mg L^−1^). As mentioned above, a reduction in marine microgel size induced by various specific conditions could lead to a decrease in the downward flux of nutrients and organic carbon, thereby disturbing the organic carbon cycle and biological pump.

Chiu et al. [52] demonstrated that the application of the dispersant Corexit (used to disrupt oil spills) can inhibit EPS aggregation and/or disperse pre-existing microgels in laboratory studies. To represent potential situations during oil spills, a water-accommodated fraction (WAF) of oil and a chemical enhanced WAF (CEWAF) were prepared by mixing oil and dispersant in artificial seawater. It was found that CEWAF can enhance EPS aggregation, with more aggregates accumulating at the air–water interface. While more hydrophobic EPS forms (higher P/C ratio) showed a high resistance to Corexit dispersion, hydrophilic EPS (lower P/C ratio) dispersed more readily when the dispersant Corexit was added, thereby suggesting that P/C ratio plays an important role in determining the stability of microgels in the presence of dispersants. In addition, Shiu et al. [31] showed a negative correlation between P/C ratio and the relative amount of extracellular DNA in EPS, indicating that a higher cellular stress level when exposed to pollutants (WAF and CEWAF) is associated with EPS of higher P/C ratios, resulting in a lower concentration of DNA. This suggests that marine microbes can actively modify their EPS release and composition in response to oil and Corexit treatments.

## 4. Incorporation of Oil and Other Hydrophobic Pollutants into Gel-Forming EPS and Marine Aggregates

Much has been written on the role of EPS-containing aggregates (‘Marine Snow’) in accomodating oil and forming ‘Marine Oil Snow’, MOS [53,54,55,56,57]. Even though EPS gels are normally hydrophilic on the outside, and they hide hydrophobic entities of mostly proteins in their interiors, hydrophobic pollutants such as oil can be accommodated well within gels upon unfolding of the proteins. EPS were found to be crucial to the formation of marine oil snow (MOS), which can form in the presence and absence of Corexit [27]. Using a radiocarbon mass balance or ^13^C-NMR quantification after a dichloromethane extraction, it was found that the presence of dispersants enhanced the amounts of protein and oil being incorporated into oil-carrying aggregates, yet slowed the sedimentation efficiency of the MOS [24]. EPS with higher P/C ratios (i.e., greater hydrophobicity) tended to facilitate the incorporation of oil and/or Corexit, and the formation of oil-carrying aggregates. When not enough mineral matter is present, colloidal aggregates can become less able to sink due to the lowered density caused by petroleum components. These observations and assessments were confirmed in subsequent mesocosm experiments that simulated both near-shore and off-shore conditions, resulting in significant relationships between the P/C ratio in aggregates/colloids and the percentage of petrocarbon incorporation into these phases regardless of conditions [34]. The P/C ratio of EPS in both the aggregate and the colloidal fraction was thus a key factor for regulating the oil contribution to the sinking aggregates. These studies also pointed out the necessity to consider more closely the presence of a mineral phase, as ballast, to overcome the buoyancy effects of oil in the oil-carrying EPS aggregates.

EPS (as the sum of individually determined proteins and carbohydrates), extracted by EDTA from the surface-attached fraction of particles in mesocosm experiments (with and without oil), correlated well with TEP [26], supporting the use of EPS as a surrogate for TEP measurements in experiments in the presence of Corexit, where TEP cannot be determined, due to analytical interference.

The water solubility of other hydrophobic pollutants such as dioxins, and PAHs, which are normally only sparingly soluble, can be greatly enhanced by their association with ‘dissolved’ organic carbon (operationally defined as passing through a 0.7 or 0.5 µm filter), which contains natural colloidal, macromolecular organic matter (operationally defined as the fraction retained by an ultrafilter of 1 or 10 kDa pore size, and passing through a 0.5 or 0.7 µm filter) composed of EPS and humic substances ([58]). For example, empirical relationships describing the binding of hydrophobic organic compounds to sedimentary (K_d_) and colloidal matter (K_c_) have been proposed and experimentally verified. The reader is referred to numerous reviews on this subject, e.g., Schwarzenbach et al. [44]. While this solubility enhancement is important for transport in more turbulent aquatic systems, in water-submerged waste disposal sites, it has been found, using state-of-the art techniques, that the truly dissolved (≤1 kDa fraction) concentration of dioxins in a waste pit was even lower than predicted from K_ow_ and BC values [45].

## 5. Specific and Nonspecific Interactions of Marine Gels with Metal Ions

The various interactions of metal ions with natural macromolecular organic molecules were reviewed in Buffle [58], Guo et al. [59], Doucet et al. [60], and Santschi et al. [32,61,62]. There are some main differences between trace metal complexation to a low-molecular-weight (LMW) ligand (e.g., citric acid) and to a high-molecular-weight (HMW) polyelectrolyte complexant, whereby the same functional group is attached to either a simple molecule or a macromolecular backbone (e.g., acid polysaccharides, macromolecular thiols, carboxylates, and proteins). LMW ligands have a small number of metal-specific functional groups, while multiple HMW ligands can be attached to different locations in the macromolecule, from where they can chelate trace metals in different ways. The nature of those ligands is relatively well recognized. The interaction between ligands and cations is generally divided into two categories depending on the hardness and softness of acids and bases. Hard acids and bases are characterized by small size, high electronegativity, and low polarizability, including A-type (e.g. Al^3+^) metals and F, O, and N. They are readily hydrated and tend to form outer-sphere complexes by ionic bonds. Soft acids and bases are characterized by relatively large size, low electronegativity, and high polarizability, including B-type metals (e.g., Ag^+^ and Hg^2+^) and S, I, and Br. They usually exist dehydrated and tend to form inner-sphere complexes by covalent bonds, which are far more stable than outer-sphere complexes.

High molecular (HMW) ligands are thus macromolecules that have a large number of surface functional groups (SFGs). They are composed of humic substances, polysaccharides, amino acids and peptides, and hydrocarbons. SFGs would be present on the outside of the biopolymer as they present themselves to the water. Due to the more complicated architecture of these biopolymers, the actual architecture can change depending on conditions (e.g., pH, redox, and salinity), and micelles could form at higher concentrations of colloidal forms of NOM. Advanced reverse osmosis/electrodialysis that consistently recovers 68 ± 2% of DOC allowed the molecular-level characterization of this macromolecular fraction via various spectroscopic (including advanced NMR) techniques [63]. It was found that condensed aromatic and quaternary anomeric carbons contribute to this deep refractory DOC pool, the quaternary anomeric carbons being a newly identified and potentially important component of bio-refractory carbohydrate-like carbon. Their results support the multi-pool (e.g. 3-pool: labile, semi-labile, and refractory) conceptual model of marine DOM biogeochemistry. Therefore, the average values of chemical (stability constants for complexation, acid-base, etc.) or physical properties (e.g., residence times) are sometimes not very meaningful and are subject to biases. The secondary effects that make up such biases can be categorized into several major groups or categories.

(1)Polyfunctional properties: They have various kinds of SFGs (R-COOH, R-OH, R-SH, R-NH_2_, etc.). Those different SFGs also have different affinities to hard and soft cations ([44]. Sometimes, a metal ion is bound to more than two SFGs. There may be competition for cations between different SFGs. For example, B-type metals have stronger affinities to (S, S) > (S, N) > (N, O) > (O, O). The same SFG can have different properties depending on the types of backbone (aliphatic or aromatic) to which they bind. Finally, the geometry, such as cavities formed near SFGs, and flexibility of the organic molecules can make a significant difference to the stability of the complexation. These kinds of steric factors are controlled by ionic strength and pH in bulk solution.(2)Conformational changes: Depending on the hydration/dehydration processes, hydrogen bonds between hydrated cations and SFGs, or metallic bridges, and the conformation of the macromolecules can form aggregates or gels. The hydration water has a different structure from that of water in the bulk solution, and it makes the stability different [58,64]. The nature of a SFG in both LMW ligand and HMW macromolecules is similar. However, the fate of the same SFG may differ depending on the fate of particles and dissolved solutes.(3)Polyelectric properties: HMW macromolecules have SFGs (e.g., R-COOH) that protonate at low pH and deprotonate at high pH. When they deprotonate under basic conditions, negatively charged SFGs repulse one another. This process creates an electric field and causes more energy needed to remove protons from SFGs, eventually increasing the pK_a_. The formation of electric fields depends on the proportion of protonated sites. This indicates that the degree of protonation or deprotonation is not solely controlled by pH in bulk solution, but also by the near-field interactions between potential ligands.(4)Binding heterogeneity effects, with binding constants becoming a function of the metal ion-to-surface site ratio [58], occur because the strongest ligands are present at the lowest concentrations, while weaker ligands are present at higher concentrations. This necessitates experimental assessments at ambient concentrations of metals and ligands, or at least use the proper ratios.(5)‘Particle concentration effects’ on kinetic constants (ki) and particle–water partition coefficients (K_d_) are a consequence of incomplete separation between particles and solution and colloids, as there often are strong metal complexing macromolecular ligands in the 0.45 µm filter-passing fraction. This effect causes experimentally determined K_d_ and k_i_ values to become a function of particle (C_p_) concentration. This ‘particle concentration effect’ on kinetic constants (k_i_) and particle–water partition coefficients (K_d_) was demonstrated using, as an example, thorium ions in the ocean, that is, the Brownian pumping model of Honeyman and Santschi [65].

Both humic substances [66] and EPS [67] can be considered a heterogeneous Donnan gel phase, similar to the situation in mucus [68]. Donnan equilibria can dominate the exchange of cations and anions across EPS gels surrounding microbial cells. For example, the Donnan mechanism affects mucin release [68] and mucus hydration [69], the swelling of exocytosed polymer-gels in *Phaeocystis pouchetii* [67], and the cation exchange membrane behavior of EPS in salt-adapted granular sludge [70]. Furthermore, toxic effects in saline environments on microbial consortia can be alleviated by the selective binding of cations to negatively charged EPS surrounding their cells, which prevents their diffusion into the deeper parts of the biofilm. The toxic effects of metal cations have been explained by various mechanisms, i.e., their ability to replace metallic enzyme cofactors, thereby disrupting the biological function of these cofactors, and the induction of redox reactions with cellular thiols, provoking Fenton-type reactions that produce reactive oxygen species and by interference with membrane transport processes [70].

When macromolecules form gels, as in the case of EPS, there will be other nonspecific interactions during crosslinking. For example, the crosslinking ability of counter-ions and the stability of the resulting networks increase with the second power of the valence electrons [71]. As an example, Fe^3+^ or Al^3+^ should be able to cross-link dissolved organic matter (DOM) at a fraction of the concentration that Ca^2+^ does. Furthermore, the degree of interaction for trivalent metal ions is higher as compared to that for divalent metal ions at physiological pH (pH ∼ 7.0) [72]). Moreover, organic polycations including spermine or spermidine, the condensing peptides of nucleic acids released from dead cells, have four cationic sites and are found in seawater at nanomolar concentrations [73]. They could be very powerful DOM crosslinkers even at submicromolar concentrations. Other polycations such as porins released by bacteria could also be interesting candidates to evaluate. 

Felz et al. [74] recently reviewed evidence of how metal ions impact structural EPS hydrogels from aerobic granular sludge. They reported that structural EPS contain alginate hydrogels, but the two are not the same. Structural EPS are more protein-rich, and their gel forming ability, stiffness determined by the Young’s modulus, and binding ability are better in the presence of transition metals (rather than alkaline earth metals) than for alginates. They also showed that structural EPS are highly complex, and they have different gelling mechanisms than the acidic polysaccharides alginate, polygalacturonic acid, and kappa carrageenan. In addition, the structural EPS hydrogels show strong integrity toward the chelating reagent EDTA.

It was clearly demonstrated that riverine and marine DOM polymers have the capacity for scavenging selected heavy metals via aggregation processes [75]. The presence of many anionic functional groups on the surface of polymers may provide cation exchange sites for complexing heavy or trace metals [76]. The highest binding affinity in three selected metals was for Cu ions, followed by Ni and Mn ions. The affinity trend is in agreement with the Irving–Williams Series (Cu > Ni > Mn) and other marine colloidal studies. For example, a higher level of colloidal Cu than colloidal Ni was found in coastal areas such as the Danshuei River estuary, Taiwan [77]; the San Francisco Bay estuary, USA [78]; and Narragansett Bay, Rhode Island [79]. This may indicate that gels binding with metals are generally affected by ligand interaction and types of polymers [75].

## 6. Gel Interactions with Nanoparticles

Studies have shown that the secretion of EPS from microbial cells is significantly affected by surrounding environmental conditions. For example, EPS with higher protein-to-carbohydrate (P/C) ratios are induced by unfavorable growth conditions, including nutrient limitation, toxins (nanoparticles, oil, and dispersant), and light exposure [16,35]. EPS with high P/C ratios are more hydrophobic and sticky, and are thus able to physically block or chemically ‘quench’ the hazardous agents [80], resulting in the lower effective concentration of toxins entering microbial cells. Furthermore, protein-rich EPS can potentially facilitate faster assembly rates of marine aggregates and alter their aggregation sizesv. The interactions between microbe growth/survival and critical characteristics of EPS (P/C ratio) in the presence of micro- and nano-plastics have received little attention. Therefore, understanding the complex biochemical interactions between three key components (microbes, nanoparticles, and EPS) during nanoparticle (NP) exposure is important to elucidate the fate of NPs, e.g., plastics, especially in their aggregation and scavenging processes in marine environments. 

## 7. Gel Interactions with Micro- and Nano-Plastics

Microplastics, as with many other micro- and nanoparticles, are rapidly covered by biofilms that then further interact with marine biogenic particles [81] to marine plastic snow (MPS). An example of nano-plastic particles enmeshed in EPS and phytoplankton cells, i.e., MPS, is given in Figure 2.

The first step is the formation of microgels. In the study by Ding et al. [15], it was shown that EPS microgel formation in seawater was greatly accelerated by small amounts of amphiphilic EPS or nano-plastics. Later, Chen et al. [16] showed that phytoplankton EPS microgel formation is also accelerated by nano-plastics in seawater, but to a lesser extent. In both cases, it was demonstrated that hydrophobic interactions dominated, which were not affected by EDTA additions, in contrast to hydrophilic interactions. Phytoplankton EPS microgel formation is greatly accelerated by nano-plastics in seawater, likely due to a higher protein content of the EPS produced [19,20]. Furthermore, EPS microgel formation from DOM was greatly accelerated by nano-plastics in different river and lake waters, as well as seawater [19]. Patches of algal cells with 1 μm polystyrene micro-particles encased in an EPS matrix have been observed [19], supporting the hypothesis that plastics would be incoporated into the phytoplankton EPS matrix to form aggregates (marine plastic snow). The sinking route of marine aggregates can scavenge micro- and nano-plastics, which may explain why the negative mass balance of entering vs. measured marine plastics in the surface waters is still lower than expected [15]. EPS effects should be considered in models for predicting and understanding the fate and transport of marine plastic debris.

## 8. Gel Interactions with Organisms

Gel interactions with marine organisms are quite diverse. For example, Haye et al. [15] showed for the first time that filter-feeding organisms effectively filter suspended particles, as well as negatively charged colloidal-sized nanoplastics (0.04 µm), at ambient (1 mg/L) concentrations, most likely as gel-like marine plastic snow. In addition, these authors showed that the chemical composition of EPS controls the uptake of metal ions from the water by oysters. For example, alginate gels greatly ameliorated the metal ion uptake, while carrageenans did not, when compared to ambient colloidal-sized EPS from Galveston Bay. Finally, biofilms composed of EPS gels growing on plastics are reservoirs for antibiotic and metal resistance genes in marine environments. Additionally, marine organisms would be unable to discriminate between target food sources and organic aggregates. These self-assembled microgels concentrate organic and inorganic materials (as above sections), and thus, the accumulation may affect the exposure of higher-trophic-level predators such as zooplankton, invertebrates, filter-feeding fish, and even humans [82,83].

## 9. Conclusions

Marine gels (analytically determined as gels, EPS or TEP) are ubiquitous, as they form from amphiphilic microbial exudates of macromolecular exopolymeric substances in aquatic systems, and they provide a physical barrier surrounding the microbial cells, mediating the transport of hydrophilic and hydrophobic compounds to and from the cells. They are mostly composed of proteins and polysaccharides, whose relative contribution can change in response to environmental conditions, e.g., the presence of pollutants. These gels, because of their amphiphilic nature, can strongly interact with ionic and nonionic constituents in various ways, e.g., through both hydrophilic and hydrophobic interactions that facilitate certain physical, chemical, or biological pathways.

## 10. Open Questions

How do marine snow and macrogels of millimeter and centimeter size form from microgels of 5 µm size?—A good start to answer this question has been made by Buffle et al. [84] in their three-colloidal component approach. In their model, they propose that large aggregates form through floc formation by crosslinking many of the individual polysaccharide-rich fibrils. However, more work needs to be carried out to understand the complex physical, chemical, and biological interactions that lead to floc and aggregate formation.

How do marine gels made from EPS respond to changes in environmental conditions, e.g., temperature, pH, redox, nutrient, ionic composition, and strength?—While global-change-related increases in temperature and hydrogen ion concentrations had been addressed by Chen et al. [48], other relevant changes in environmental conditions had not been properly addressed.

## Figures and Tables

**Figure 1 gels-07-00083-f001:**
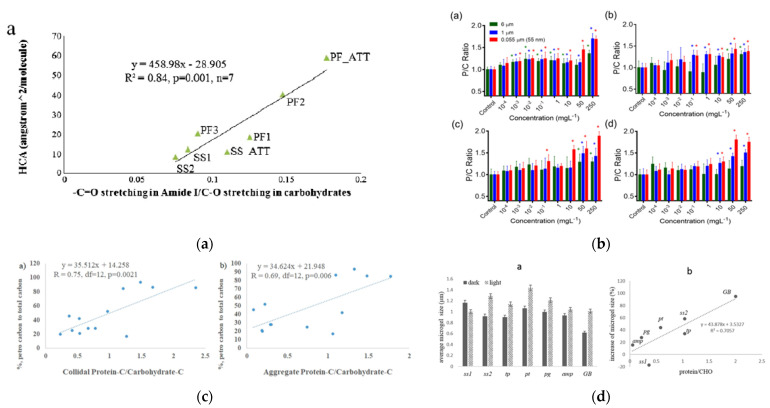
(**a**) Examples for the relative hydrophobicity of EPS that increases with P/C ratio ([20], with permission of the publisher), (**b**) the relationship between nanoplastics concentration and the size-dependent induction of EPS with higher P/C ratio ([19], with permission of the publisher), (**c**) the relationships of % petro-carbon to total carbon in colloidal or sinking aggregates that increase with the P/C ratio of EPS ([34], with permission of the publisher), and (**d**) the microgel size increase due to light-induced ROS chemical crosslinking of proteins in EPS that scale with their P/C ratio ([35], with permission of the publisher).

**Figure 2 gels-07-00083-f002:**
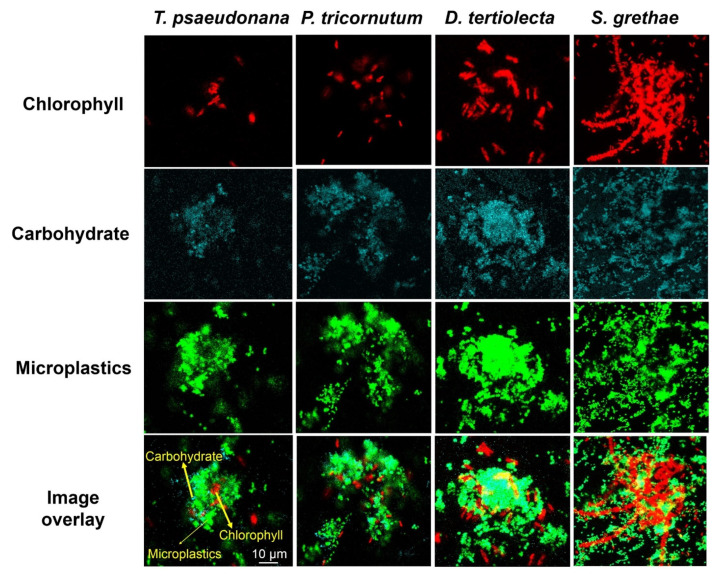
Marine Plastic Snow ([19], with permission from the publisher).

**Table 1 gels-07-00083-t001:** Terminology used in this paper.

NOM	natural organic matter
DOM	dissolved organic matter (i.e., passing a filter of about 0.5 µm pore size)
DOC	dissolved organic carbon (i.e., passing a filter of about 0.5 µm pore size)
EPS	exopolymeric substances, found in the colloidal or particulate fraction
TEP	transparent exoplymeric particles, operationally determined
Gels	a type of soft matter that is operationally determined in aquatic systems
HMW	high molecular weight (relative term, usually more than 1 kDa)
LMW	low molecular weight (relative term, usually less than 1 kDa
SFG	surface functional group
DLS	dynamic light scattering
FTIR	fourier transform infrared spectroscopy

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
