# Peer review of "Marine Gel Interactions with Hydrophilic and Hydrophobic Pollutants"

_gels, 2021, doi:10.3390/gels7030083_

Round 1

Reviewer 1 Report

The manuscript is a review of the literature on the interaction of marine gels with hydrophilic and hydrophobis pollutants. The manuscript describes important chemical properties of marine gels especially in relation to reactions with other charged and uncharged substances, in particular pollutants. The manuscript is well written and covers essential aspects of the subject. However, it is not a balanced review of the current literature, but mainly summarises the authors' own work to date.

Minor comments:

Introduction:

Line 26 ..and nanoparticles, including microplastics and nanoplastics Microplastics do not belong to the nanoparticles pool

line 39/40: Add reference

lines 47-62: Add CSP as major gel particle type

line 60: Add reference

line 77: P/C ratio, consider using another abbreviation to avoid confusion with the element ratio

Chapter 4 Incorporation of Oil and ...

There is relevant work form other goups on oil pollutants and EPS/ Marine gels (e.g. Passow lab, Mari lab). In general, consider citing relevant work of other working groups on the topic (e.g. U. Passow, X. Mari, A. Engel, D. Thornton, O. Wurl, C. Cunha)

Author Response

Response to Reviewer 1

The manuscript is a review of the literature on the interaction of marine gels with hydrophilic and hydrophobis pollutants. The manuscript describes important chemical properties of marine gels especially in relation to reactions with other charged and uncharged substances, in particular pollutants. The manuscript is well written and covers essential aspects of the subject. However, it is not a balanced review of the current literature, but mainly summarises the authors' own work to date.

Minor comments:

Introduction:

Line 26 ..and nanoparticles, including microplastics and nanoplastics Microplastics do not belong to the nanoparticles pool

Response: deleted microplastics

line 39/40: Add reference

Response: added

lines 47-62: Add CSP as major gel particle type

Response: added, with reference

line 60: Add reference

Response: added

line 77: P/C ratio, consider using another abbreviation to avoid confusion with the element ratio

Response: Since we already published this ratio a number of times, we would like to keep it to be in line with previous abbreviations.

Chapter 4 Incorporation of Oil and ...

There is relevant work form other goups on oil pollutants and EPS/ Marine gels (e.g. Passow lab, Mari lab). In general, consider citing relevant work of other working groups on the topic (e.g. U. Passow, X. Mari, A. Engel, D. Thornton, O. Wurl, C. Cunha)

Response: We agree with this reviewer that more references on EPS interactions with oil need to be added. We added now more references from these authors if they published on EPS/oil, or on EPS alone.

Reviewer 2 Report

This manuscript is a review work on the subject of marine exopolymeric substances, EPS, forming gels and the interactions of these EPS-based gels with various pollutants. This is definitely a compelling topic: the polymer science as well as marine science communities ought to focus in this direction as a major class of pollutants are man-made plastics/polymerics (whatever size category) and as the pollutants change behavior of marine gels that may influence carbon cycle of the planet.

As a review, the work does not represent a significant contribution to the field -  but it serves the educational function and brings up an important topic. 

Thus, I consider this expert-written review very much useful.

I think that the text is ready to be published in Gels as presented. Indeed, after few technical corrections that will be listed below.

However, I have few suggestions for the authors to be considered and perhaps few changes done before final submission just to make the work more complete and more speaking to polymer and materials science audience.

 1/ Technical and language

Check and explain all abbreviations:

line 17: instead EP should be EPS

line 113: explain EDTA

remove lines 146 – 161, the text doubles

line 227: readily hydrate should be readily hydrated

lines 258-259...the hydration water has a different structure from that in the bulk...this demands source/citation

line 261: SFG may different should be SFG may differ

line 306:  DOM crosslinker should be DOM crosslinkers

line 323 higher colloidal Cu / higher Ni should higher level of colloidal Cu / level of Ni

2/ Amendments, clarifications

It would be helpful if the basic terms are clearly explained in the intro, such as relations between EPS–TEM and their relation to NOM, DOM – DOC, HMW etc. Maybe a scheme? Is there some hierarchy or how these entities relate to each other? Not least, some quantitative info on for example the total amounts in the sea or similar would be interesting.

Are EPS adjacent to organisms or even exerting some functions for the organisms such as binders, extracellular matrix, food or are just freely floating in the sea?

Are there some degradation processes of EPS-gels?

What is the relation between EPS gels and TEP, are these terms synonymous?

What are characteristic properties of EPS-based gels? (rheology, morphology, specific gravity,...etc)

Line 50: what is the principle of determination of EPS in filter-passing fraction? Will appreciate brief explanation without the need to look up the source. If you say gel content in filter-passing fraction, it means that some gel does not pass through the filter and more gel is present in the fraction that vent through the filter?

The P/C ratio seems to be crucial and it relates to many properties. It is discussed in several sections of the article. A scheme of effects along P/S value drawn for example on linear axis would make reading more digestible.

Line 101: what do you mean by: Due to the solubility limitation of water, there are no hydrophobic EPS?

Lines 198-201 : the text is not understandable or too technical for not a deeply involved expert. It says that EPS correlated well with TEP but it also says that TEP could not be determined.  

 Lines 202 and farther: I can’t agree with the statement that solubility of dioxins and other in water...was enhanced upon their association with DOM: the solubility of the substances remains the same but the solubility of their associates with other substances is different. Then, it is still a question in what form the dioxins (and other pollutants) in associates present when “dissolved”  - could be hydrophobically interacting with the part of DOM carrier, so technically not dissolved but rather dispersed and quite likely sags down as the ECM-gel snow (for example).

Line 232: HMW ligands are macromolecules...with surface functional groups. Does this imply that HMW are present in form of some micelles or domains of certain shape where we can define a surface? Otherwise I can’t conceptualize a surface of macromolecule...Pls explain.

Author Response

Reviewer 2

Comments and Suggestions for Authors

This manuscript is a review work on the subject of marine exopolymeric substances, EPS, forming gels and the interactions of these EPS-based gels with various pollutants. This is definitely a compelling topic: the polymer science as well as marine science communities ought to focus in this direction as a major class of pollutants are man-made plastics/polymerics (whatever size category) and as the pollutants change behavior of marine gels that may influence carbon cycle of the planet.

As a review, the work does not represent a significant contribution to the field -  but it serves the educational function and brings up an important topic. 

Thus, I consider this expert-written review very much useful.

I think that the text is ready to be published in Gels as presented. Indeed, after few technical corrections that will be listed below.

However, I have few suggestions for the authors to be considered and perhaps few changes done before final submission just to make the work more complete and more speaking to polymer and materials science audience.

 1/ Technical and language

Check and explain all abbreviations:

line 17: instead EP should be EPS

Response: done

Response:

line 113: explain EDTA

Response: done

remove lines 146 – 161, the text doubles

Response: doubles eliminated

line 227: readily hydrate should be readily hydrated

Response: done

lines 258-259...the hydration water has a different structure from that in the bulk...this demands source/citation

Response: added

line 261: SFG may different should be SFG may differ

Response: done

line 306:  DOM crosslinker should be DOM crosslinkers

Response: done

line 323 higher colloidal Cu / higher Ni should higher level of colloidal Cu / level of Ni

Response: rewritten

2/ Amendments, clarifications

It would be helpful if the basic terms are clearly explained in the intro, such as relations between EPS–TEM and their relation to NOM, DOM – DOC, HMW etc. Maybe a scheme? Is there some hierarchy or how these entities relate to each other? Not least, some quantitative info on for example the total amounts in the sea or similar would be interesting.

Response: we agree with this reviewer, and provide a scheme or table for the nomenclature that was used here.

Table 1. Terminology used in this paper.

NOM = natural organic matter

DOM = dissolved organic matter (i.e., passing a filter of about 0.5 µm pore size)

DOC = dissolved organic carbon (i.e., passing a filter of about 0.5 µm pore size)

EPS = exopolymeric substances, found in the colloidal or particulate fraction

TEP = transparent exoplymeric substances, operationally determined

Gels = a type of soft matter, that is operationally determined in aquatic systems

HMW = high molecular weight (relative term, usually more than 1kDa)

LMW = low molecular weight (relative term, usually less than 1kDa

SFG = surface functional group

Are EPS adjacent to organisms or even exerting some functions for the organisms such as binders, extracellular matrix, food or are just freely floating in the sea?

Response: We now explain this some more. There are attached EPS biopolymers which are generally richer in proteins, and free-floating EPS that are usually protein poorer.

Are there some degradation processes of EPS-gels?

Response: Yes, we added some more explanations. The plankton – EPS system is a dynamic system, whereby phytoplankton and bacteria form a synergistic relationship in the phycosphere. Phytoplankton secrete photosynthesized carbohydrates/polysaccharides, and associated bacteria degrade some of this material and make available other compounds such as Vitamin B and hydroxamate siderophores to phytoplankton.

What is the relation between EPS gels and TEP, are these terms synonymous?

Response: These terms are not synonymous, and we added some sections to further clarify this (see also Xu, C., Chin, W.-C., Lin, P., Chen, H.M., Lin, P., Chiu, M.-C., Waggoner, D.C., Xing, W., Sun, L., Schwehr, K.A., Hatcher, P.G., Quigg, A., Santschi, P.H., 2019. Marine Gels, Extracellular Polymeric Substances (EPS) and Transparent Exopolymeric Particles (TEP) in natural seawater and seawater contaminated with a water accommodated fraction of Macondo oil surrogate. Marine Chemistry; https://doi.org/10.1016/j.marchem.2019.103667. )

Difference between EPS, TEP, and GELS

Extracellular polymeric substances (EPS) are colloidal-sized macromolecules that are secreted by bacteria and phytoplankton into the water. They are predominantly composed of polysaccharides and proteins, but also contain various other components such as DNA and glycolipids (Bhaskar and Bhosle, 2005; Bhaskar et al., 2005; Quigg et al., 2016; Xu et al., 2009). Some EPS contain high concentrations of uronic acids, which confer polyanionic attributes (Decho and Gutierrez, 2017). In addition, the presence of amino acid and peptide moieties render amphiphilic characteristics to these macromolecules. These surface active properties result in important marine biogeochemical processes, including microbial adhesion, biofilm formation, and trace metal-radionuclide-nutrient binding contributing to their cycling, as well as emulsification and biodegradation of petroleum components, etc., thus greatly impacting whole ecosystems (Decho and Gutierrez, 2017).

Transparent Exopolymeric Particles, TEP, are commonly considered to be precursers of EPS. They are ubiquitously present in marine and fresh water systems yet “non-visible” under the microscope unless they are stained (e.g., Alcian blue; (Passow and Alldredge, 1995)). TEP are primarily assessed as acidic polysaccharides (Passow, 2002a; Passow, 2002b). EPS and TEP do not refer to exactly the same materials: TEP “are exopolymers, but not all exopolymeric substances occur as TEP or can form TEP” (Passow, 2002b). EPS forming microbial biofilms have shown to be gels (Bar-Zeev et al., 2012; Chin et al., 1998; Dohnalkova et al., 2011; Verdugo, 2012). However, TEP are not strictly considered to be gels, since their formation relies on the coagulation theory, not on in-termolecular energies and assembly processes as for gel formation. Nevertheless, these terms (EPS, TEP, gels)  are often used interchangeably, as in the case of biofilm formation and biofouling (Berman, 2011).

Gels are conceptually considered a type of soft matter (de Gennes, 1992) and are well-defined (Chin et al., 1998; Verdugo, 2012). However, microgel concentrations are operationally determined using flow cytometry, after staining with chlortetracycline, expressed as total organic carbon concentration (Ding et al., 2007; Verdugo et al., 2008), with the kinetics of gel formation determined using Dynamic Light Scattering over hours to days (Chin et al., 1998). Coomassie stainable particles (CSP), which are pro-tein-containing particles and can be stained with Coomassie Brilliant Blue, are another type of gel-like particle, visualized by Long and Azam (1996) that has been described in seawater, freshwater and phytoplankton cultures. TEP and CSP could be discrete particles, or subunits of the same particles (Thornton, 2018).

What are characteristic properties of EPS-based gels? (rheology, morphology, specific gravity,...etc)

Response: All these physical properties depend on various physical and chemical factors, like cross-linker density, cross-liner types, polymer length, pH, types of polymers, temperature, degree of swelling or temperature.  Unfortunately, to the best of our knowledge, there is no available direct measurement for these properties for natural EPS gels in the literature.  However, several studies on alginate (or other purified EPS) are available in the literature that might provide some rough assessments.  Mechanical and rheological (viscoelastic) properties of alginate gels were shown to be dependent on the cross-linker type, density, ionic conditions, gelling temperatures, or EPS concentrations [1-4].  Specific gravity of a typical synthetic hydrogel (PVA gels) is around 1.05 [5]. For EPS sludges, the density has been reported to be around 1.004-1.048 (g/ml) [6].  For alginate gels depending on the gelling conditions, the value can vary from 1.03 to 1.12 (g/ml) [1].  However, these specific gravity measurements were conducted in non-seawater conditions (in lower salt conditions). Please note the specific gravity of seawater is around 1.025. The specific gravity of EPS gels in seawater might shift from the measurements in non-seawater environments. The appearance and sizes of marine EPS gels are highly heterogeneous. No typical or characteristic morphology/shape has been concluded.  The size of EPS gels in seawater can range from sub-micrometers to millimeters, even several centimeters.  

EPS is highly heterogeneous mixtures of biopolymers from various microbes in seawater and generally associated with different type of particles (anthropogenic, minerals or biological debris), these physical properties (rheology, morphology, or specific gravity) of natural EPS gels are usually complex and highly variable to determine or generalize.

Line 50: what is the principle of determination of EPS in filter-passing fraction? Will appreciate brief explanation without the need to look up the source. If you say gel content in filter-passing fraction, it means that some gel does not pass through the filter and more gel is present in the fraction that vent through the filter?

Response: We added a new section to clarify this point

The P/C ratio seems to be crucial and it relates to many properties. It is discussed in several sections of the article. A scheme of effects along P/S value drawn for example on linear axis would make reading more digestible.

Response: We added a new section on the P/C, where we added a new scheme, as a figure.

Line 101: what do you mean by: Due to the solubility limitation of water, there are no hydrophobic EPS?

Response: As explained before, proteins have hydrophobic moieties that are normally hidden in their interior, but can be exposed by unfolding. Hydrophobic interactions then can happen after encountering a hydrophobic substance. We now make this clear.

Lines 198-201 : the text is not understandable or too technical for not a deeply involved expert. It says that EPS correlated well with TEP but it also says that TEP could not be determined.  

Response: We clarified this point now, as TEP only correlated with EPS when no corexit was present. TEP could not be determined in the presence of corexit, as it interfered in the analysis.

 Lines 202 and farther: I can’t agree with the statement that solubility of dioxins and other in water...was enhanced upon their association with DOM: the solubility of the substances remains the same but the solubility of their associates with other substances is different. Then, it is still a question in what form the dioxins (and other pollutants) in associates present when “dissolved”  - could be hydrophobically interacting with the part of DOM carrier, so technically not dissolved but rather dispersed and quite likely sags down as the ECM-gel snow (for example).

Response: We clarified this now. Often, ‘dissolved’ is used as meaning present in the <0.5µm filterpassing fraction. So, this reviewer is correct, it is technically not dissolved but found in the colloidal fraction. This is now clarified.

Line 232: HMW ligands are macromolecules...with surface functional groups. Does this imply that HMW are present in form of some micelles or domains of certain shape where we can define a surface? Otherwise I can’t conceptualize a surface of macromolecule...Pls explain.

Response: We clarified this now. Surface functional groups would be present on the outside of the biopolymer as it presents itself to the water. Due to the more complicated architecture, this architecture can change, and micelles could form at higher concentrations of colloidal forms of NOM.

References

  1. Jeong, C.; Kim, S.; Lee, C.; Cho, S.; Kim, S.-B. Changes in the physical properties of calcium alginate gel beads under a wide range of gelation temperature conditions. Foods 2020, 9, 180.
  2. Lee, K.Y.; Rowley, J.A.; Eiselt, P.; Moy, E.M.; Bouhadir, K.H.; Mooney, D.J. Controlling mechanical and swelling properties of alginate hydrogels independently by cross-linker type and cross-linking density. Macromolecules 2000, 33, 4291-4294.
  3. Lotti, T.; Carretti, E.; Berti, D.; Montis, C.; Del Buffa, S.; Lubello, C.; Feng, C.; Malpei, F. Hydrogels formed by anammox extracellular polymeric substances: structural and mechanical insights. Scientific reports 2019, 9, 1-9.
  4. Kakita, H.; Kamishima, H. Some properties of alginate gels derived from algal sodium alginate. In Proceedings of the Nineteenth International Seaweed Symposium, 2008; pp. 93-99.
  5. Sajjan, A.; Banapurmath, N.; Tapaskar, R.; Patil, S.; Kalahal, P.; Shettar, A. Preparation of polymer electrolyte hydrogels using poly (vinyl alcohol) and tetraethylorthosilicate for battery applications. In Proceedings of the IOP Conference Series: Materials Science and Engineering, 2018; p. 012078.
  6. Feng, C.; Lotti, T.; Canziani, R.; Lin, Y.; Tagliabue, C.; Malpei, F. Extracellular biopolymers recovered as raw biomaterials from waste granular sludge and potential applications: A critical review. Science of the Total Environment 2020, 142051.

Round 2

Reviewer 1 Report

The authors have done a good job by including more relevant research aspects in their manuscript. This is a nice and comprehensive compilation of studies on the topic of Marine Gel interactions with pollutants.